# AQP3 and AQP9—Contrary Players in Sepsis?

**DOI:** 10.3390/ijms25021209

**Published:** 2024-01-19

**Authors:** Patrick Thon, Tim Rahmel, Dominik Ziehe, Lars Palmowski, Britta Marko, Hartmuth Nowak, Alexander Wolf, Andrea Witowski, Jennifer Orlowski, Björn Ellger, Frank Wappler, Elke Schwier, Dietrich Henzler, Thomas Köhler, Alexander Zarbock, Stefan Felix Ehrentraut, Christian Putensen, Ulrich Hermann Frey, Moritz Anft, Nina Babel, Barbara Sitek, Michael Adamzik, Lars Bergmann, Matthias Unterberg, Björn Koos, Katharina Rump

**Affiliations:** 1Klinik für Anästhesiologie, Intensivmedizin und Schmerztherapie, Universitätsklinikum Knappschaftskrankenhaus Bochum, 44892 Bochum, Germany; patrick.thon@rub.de (P.T.); tim.rahmel@rub.de (T.R.); dominik.ziehe@rub.de (D.Z.); lars.palmowski@kk-bochum.de (L.P.); britta.marko@kk-bochum.de (B.M.); hartmuth.nowak@kk-bochum.de (H.N.); andrea.witowski@kk-bochum.de (A.W.); jennifer.orlowski@kk-bochum.de (J.O.); barbara.sitek@rub.de (B.S.); michael.adamzik@kk-bochum.de (M.A.); lars.bergmann@kk-bochum.de (L.B.); matthias.unterberg@kk-bochum.de (M.U.); bjoern.koos@ruhr-uni-bochum.de (B.K.); 2Center for Artificial Intelligence, Medical Informatics and Data Science, University Hospital Knappschaftskrankenhaus Bochum, 44892 Bochum, Germany; 3Klinik für Anästhesiologie, Intensivmedizin und Schmerztherapie, Klinikum Westfalen, 44309 Dortmund, Germany; bjoern.ellger@klinikum-westfalen.de; 4Department of Anesthesiology and Operative Intensive Care Medicine, University of Witten/Herdecke, Cologne Merheim Medical School, 51109 Cologne, Germany; wapplerf@kliniken-koeln.de; 5Department of Anesthesiology, Surgical Intensive Care, Emergency and Pain Medicine, Ruhr-University Bochum, Klinikum Herford, 32049 Herford, Germany; elke.schwier@klinikum-herford.de (E.S.); dietrich.henzler@klinikum-herford.de (D.H.); thomas.koehler@ruhr-uni-bochum.de (T.K.); 6Klinik für Anästhesiologie, Operative Intensivmedizin und Schmerztherapie, Universitätsklinikum Münster, 48149 Münster, Germany; zarbock@uni-muenster.de; 7Klinik für Anästhesiologie und Operative Intensivmedizin, Universitätsklinikum Bonn, 53127 Bonn, Germany; stefan.ehrentraut@ukbonn.de (S.F.E.); christian.putensen@ukbonn.de (C.P.); 8Marien Hospital Herne, Universitätsklinikum der Ruhr-Universität Bochum, 44625 Herne, Germany; ulrich.frey@elisabethgruppe.de; 9Center for Translational Medicine, Medical Clinic I, Marien Hospital Herne, University Hospital of the Ruhr-University Bochum, 44625 Herne, Germany; moritz.anft@elisabethgruppe.de (M.A.); nina.babel@elisabethgruppe.de (N.B.)

**Keywords:** aquaporins, sepsis, critical illness, AQP3, AQP9, mortality

## Abstract

Sepsis involves an immunological systemic response to a microbial pathogenic insult, leading to a cascade of interconnected biochemical, cellular, and organ–organ interaction networks. Potential drug targets can depict aquaporins, as they are involved in immunological processes. In immune cells, AQP3 and AQP9 are of special interest. In this study, we tested the hypothesis that these aquaporins are expressed in the blood cells of septic patients and impact sepsis survival. Clinical data, routine laboratory parameters, and blood samples from septic patients were analyzed on day 1 and day 8 after sepsis diagnosis. AQP expression and cytokine serum concentrations were measured. AQP3 mRNA expression increased over the duration of sepsis and was correlated with lymphocyte count. High AQP3 expression was associated with increased survival. In contrast, AQP9 expression was not altered during sepsis and was correlated with neutrophil count, and low levels of AQP9 were associated with increased survival. Furthermore, AQP9 expression was an independent risk factor for sepsis lethality. In conclusion, AQP3 and AQP9 may play contrary roles in the pathophysiology of sepsis, and these results suggest that AQP9 may be a novel drug target in sepsis and, concurrently, a valuable biomarker of the disease.

## 1. Introduction

Sepsis is a common life-threatening syndrome caused by dysfunctional host response to infection and is associated with organ dysfunction [1]. Sepsis is associated with about 20% of deaths worldwide [2] and remains one of the leading causes of death in industrialized nations [3], causing millions of deaths every year [4]. There are approximately 48.9 million global sepsis cases per year [5]. In addition, sepsis is among the most expensive conditions treated in hospitals [6]. Biomarkers for the early prediction and diagnosis of sepsis are studied extensively. As an example, the biomarker panel of procalcitonin (PCT), C-reactive protein (CRP), and serum amyloid A (SAA) seems to be predictive of sepsis [7]. The diagnosis of sepsis in critically ill patients is a challenge, as conventional infection markers are often misleading. Therefore, new technologies are increasingly being considered, such as omics-based technologies for the identification of immune profiles for the diagnosis and prognosis of sepsis. However, the technologies are currently becoming difficult to use in clinical practice due to their complexity, the amount of work involved, and the lack of standardization [8,9].

In recent years, rapid diagnosis and appropriate intensive care treatments have partly reduced the mortality rate of sepsis, but effective treatments for sepsis are still lacking [10]. In addition, the high complexity of the disease and long-term treatment lead to numerous undesirable consequences, such as mental, physical, and functional disorders [10,11]. For this reason, there is an urgent need to identify new key proteins for sepsis treatment and to further elucidate the underlying molecular mechanisms. In addition, the novel approach of patient enrichment opens up new avenues for precision medicine approaches aimed at applying immunotherapies against sepsis based on precise biomarkers and molecular mechanisms that define specific immune endotypes [12].

Aquaporins (AQPs) may represent interesting target genes for sepsis therapy [13]. AQPs are a group of 13 membrane proteins that are essential for the regulation of water and salt influx and efflux in cells. In addition, some AQPs facilitate the transport of glycerol and other small solutes, such as urea and carbon dioxide, across cell membranes [13]. AQPs are membrane channel proteins that can selectively and efficiently transport water and other small molecules [14]. Water-selective AQPs are involved in many biological functions, including transepithelial fluid transport, cell migration, brain edema, and neuroexcitation [13,14], whereas aquaglyceroporins are involved in cell proliferation, adipocyte metabolism, and epidermal water retention. Recent studies have shown that in immune systems, at least six AQPs, including AQP1, AQP3, AQP4, AQP5, AQP7, and AQP9, are expressed [13,15,16,17]. These AQPs are distributed in lymphocytes, macrophages, dendritic cells, and neutrophils, and they mediate water and glycerol transportation in these cells, which play important roles in innate and adaptive immune functions. In addition, they regulate the migration of different immune cells. AQP3 and AQP9 are of special interest, as AQP3 appears to be important for T-cell function [18] and AQP9 regulates neutrophil cell migration [19]. Their role in the immune system may have a significant impact on sepsis and influence its pathophysiology. Hence, in this study, we tested the hypotheses that these AQPs are differentially expressed in the blood of septic patients, are related to immune cell count, and impact sepsis survival.

## 2. Results

### 2.1. Baseline Characteristics of the Patients

AQP expression was quantified in the whole blood samples of 87 septic patients. At study inclusion, the median sequential organ failure assessment (SOFA) score of the cohort was 8 (IQR: 5–11). The median age was 68 years (IQR: 61–79), and 53% of the patients were male (Table 1).

### 2.2. AQP Expression in the Patients

The expressions of AQP3 and AQP9 were quantified in the whole blood samples of 87 septic patients. On the day of study inclusion, AQP9 showed higher mRNA expression in blood samples compared with AQP3 (*p* < 0.0001; Figure 1). The expression of the other aquaporins AQP1, AQP5, AQP7, AQP8, and AQP10 was lower than the expression of AQP3 and AQP9 (Appendix A).

AQP expression was analyzed over the duration of sepsis. AQP3 (*p* < 0.0001; Figure 2) expression more than doubled when comparing day 8 of sepsis with day 1, whereas AQP9 mRNA expression was not altered over the duration of sepsis (Figure 2). Regarding further aquaporins, an increase in AQP expression was observed for AQP5 (*p* = 0.0022; Appendix A).

AQP3 expression was negatively correlated with neutrophil (*p* < 0.0001, Figure 3) and leucocyte (*p* = 0.012, Figure 3) cell counts and positively correlated with lymphocyte cell count (*p* = 0.006, Figure 3).

In contrast to AQP3, AQP9 showed a strong positive correlation with neutrophil cell count (*p* = 0.0017, Figure 4) and negative correlations with lymphocytes and classical monocytes (*p* < 0.0257; Figure 4).

Regarding cytokine release in the serum of septic patients, we identified a negative correlation between AQP3 mRNA and interleukin-8 (IL-8) and negative correlations between AQP9 mRNA and interleukin-1β (IL-1β), interferon-α2 (IFN-α2), and interleukin-33 (IL-33). All these correlations were relatively weak (*p* < 0.04, Figure 5).

At day 8, the negative correlations between AQP mRNA expression and serum cytokine concentration persisted. However, only the correlation between IL-8 and AQP3 concentration (r = −0.529; *p* = 0.002) and AQP9 and IFN-α2 (r = −0.5066; *p* = 0.0136) were relatively strong and reached significance levels (Figure 6). 

In a final step, we examined whether AQP mRNA expression could be a potential risk factor for sepsis lethality. We found AQP mRNA expression to be associated with 30-day survival. Kaplan–Meier analysis was performed after determining the point of best discrimination between sepsis survivors and non-survivors using ROC analysis with the Youden index. This point was defined as the cut-off value for each AQP. As the Kaplan–Meier analysis showed, patients with an AQP3 expression that was above the cut-off value had a significantly greater chance of survival than those with a lower AQP3 expression on day 8 (82.4% vs. 43.8%, *p* = 0.017; Figure 7). In contrast, the Kaplan–Meier curve showed increased survival in patients with lower AQP9 expression (68.2% vs. 20.0%, *p* = 0.003; Figure 7). This effect was independent of age and gender (*p* = 0.003, Cox regression analysis; Table 2). The extent of this effect is reflected in the hazard ratio, which was univariately determined to be 5.59 (95% CI: 1.579–19.563, *p* = 0.008). Therefore, we found that in this cohort, elevated levels of AQP9 expression were detrimental to patient survival.

## 3. Discussion

In this study, we showed that increased AQP9 mRNA expression is an independent risk factor for 30-day lethality in sepsis. In contrast, elevated AQP3 expression in septic patients seems to be beneficial for survival. Furthermore, AQP9 appears to be the most abundantly expressed AQP in whole blood samples of septic patients, followed by AQP3. However, in contrast to AQP3, the expression of which increases over the duration of sepsis, AQP9 expression seems to be unaltered. Interestingly, AQP9 expression is positively correlated with neutrophil count and negatively correlated with lymphocyte count, whereas we found the opposite for AQP3 expression. Therefore, our results may suggest an opposing role for AQP9 and other AQPs like AQP3 in sepsis.

In order to understand the role of AQPs in sepsis, we would first like to take a brief look at the pathomechnism of sepsis. In short, the systemic activation of the immune system during sepsis results in an inflammatory response characterized by a cytokine storm with associated fever, shock, and multiple organ dysfunction. The activation of the innate immune system includes the activation of neutrophils and monocytes, which infiltrate infected tissues and acquire inflammatory macrophage and dendritic cells [20,21]. Activated neutrophils interact with platelets and are involved in dysregulated coagulation [20]; however, most bacteria are readily killed by neutrophils [20]. In addition, as mentioned above, sepsis is accompanied by a cytokine storm, and interleukin-6 IL-6), IL-8, IL-1β, IFN-α2, and IL-33 are released [21]. In the following paragraphs, we would like to discuss the relationship between AQP3 and AQP9.

To obtain a better understanding of their impact on sepsis, we have to consider the role of the particular AQPs in the immune system and in sepsis. AQP9 regulates the migration of different immune cell types and is considered to be one of the most relevant AQPs in the immune system [22]. AQP9 is expressed in neutrophils, leucocytes, dendritic cells, macrophages, and monocytes [23]. AQP9 can be found at cell edges in leukocytes and facilitates lamellipodium extension, allowing cells to migrate toward a chemoattractant gradient [24]. In our analysis, AQP9 correlates strongly with neutrophil count, indicating high AQP9 expression in these cells. AQP9 expression, mainly in neutrophils, has been described elsewhere [25], and its importance in neutrophil migration has been well described [24]. In addition, some animal studies regarding the impact of AQP9 on the pathomechnism of sepsis have been conducted. In an animal model of lipopolysaccharide (LPS)-induced endotoxemia, *Aqp9*-KO mice showed increased survival compared with wild-type mice [26]. This is consistent with our finding that reduced AQP9 mRNA expression is correlated with increased survival. Furthermore, *Aqp9*-KO mice showed reduced inflammatory nitric oxide (NO) and superoxide production. They also showedinducible NO synthase (iNOS) and cyclooxygenase-2 (COX-2) levels achieved by decreased NF-κB p65 activation [26]. In addition, a study using an AQP9 inhibitor in a mouse cecal ligation puncture (CLP)-model for sepsis indicated that administration AQP9 inhibitor attenuates the cardiac and renal dysfunction caused by sepsis, reduces the activation of the NLR family pyrin domain containing the 3 (NLRP3)-inflammasome pathway and myeloperoxidase (MPO) activity in lung tissue [27]. These results indicate that AQP9 may be a novel drug target in sepsis.

Considering our data, AQP3 may be another important player in immune cells during sepsis. AQP3 was found to be expressed in human leucocytes, lymphocytes, dendritic cells, and T-cells [23,28]. Our data indicate that AQP3 expression is correlated with lymphocytes. It has been previously described that AQP3 is expressed in cytotoxic T-lymphocytes [29] and required for T-cell function [28]. In addition, it has been found that mouse resident peritoneal macrophages express AQP3 in a plasma membrane pattern, and that *Aqp3*-KO mice have greater mortality than wild-type mice in a murine model of bacterial peritonitis [30]. Here, we find a commonality with our results, where patients with reduced AQP3 mRNA expression have increased mortality. This is in contrast to a recent study that showed that AQP3 participates in the regulation of pulmonary vascular permeability after sepsis and that the antioxidant Ss-31 has a protective effect on pulmonary vascular permeability by downregulating the expression of AQP3 and inhibiting reactive oxygen species (ROS) production, which led to increased survival in a CLP model [31]. In addition to lung function [32], AQP3 might also play a role in the stroma [33] and the kidneys [34] during sepsis.

Since cytokines are important mediators of innate immune responses, we also examined whether AQPs are correlated with cytokine concentration.

Our study demonstrates that AQP3 is only correlated with IL-8 concentration. An AQP3-dependent IL-8 cytokine release has been reported previously [35]; however, correlations with other cytokines like IL-6 and interleukin-18 (IL-18) were not detectable in our analysis. It is known that AQP3 is involved in NF-κB signaling and NLRP3-inflammasome activation, and the downregulated expression of AQP3 inhibits the production of cytokines such as IL-6, tumor necrosis factor α (TNF-α), and IL-1β [36,37]. Hence, AQP3 seems to be an important player in immune response and might influence cytokine signaling pathways. However, the precise mechanisms through which AQP3 affects the production of inflammatory cytokines in sepsis have to be elucidated.

AQP9 expression is correlated with IL-1β, IFN-α2, and IL-33. The findings of a previous study, in which IL-1β induces AQP9 expression, cannot be directly confirmed by our results [38]. IL-33 seems to be important for eosinophil granulocyte function in sepsis [39]. In addition, it is known that IL-33 is a potent activator of type 2 Innate Lymphoid Cells (ILC2), which results in enhanced production of the key effector cytokines IL-5 and IL-13, and that IL-33 plays a major role in local inflammatory changes in the lung, early after the onset of sepsis [40]. Hence, it would be interesting to study the ILC2 population, which was not carried out in this study. However, AQP9 expression seems to be more prominent in neutrophil cells [39,41]. Regarding IL-8, no association with AQP3 and AQP9 expression have been shown; however, IL-8 is reduced after LPS stimulation when AQP1 is overexpressed [42]. In addition, there are no studies about a correlation between AQP9 and interferon signaling. Therefore, we can assume that the correlation we found between cytokine secretion and AQP expression might be caused by variations in cell counts and due to a different cytokine release by the different cell types.

The main finding of our study is the correlation between AQP3 and AQP9 expression and sepsis survival at day 8, suggesting AQP modulation might be a promising therapeutical approach during sepsis. AQPs may, therefore, represent an interesting drug target in sepsis, which could be of interest not only in the early phase of sepsis but also in its later course. Our data indicate that AQP9 is inhibited during sepsis. Specific inhibitors for particular aquaporins are not easy to identify [13], as there are several human AQP homologs, many with a wide tissue distribution. Recently, a specific inhibitor for AQP9 was identified [43,44]. The use of specific AQP modulators is particularly important, as high AQP3 expression was beneficial in our study. The upregulation of AQPs can be mediated by different drugs. As an example, our group showed an upregulation of AQP1 by the β2 adrenoreceptor agonist terbutaline, and AQP3 is upregulated by ambroxol [44,45]. Another promising inductor of AQP3 might be tanshinone IIA, which can increase AQP3 protein expression [46].

Taken together, these results indicate that AQP9 or AQP3 could be novel drug targets in polymicrobial sepsis and valuable biomarkers of this worrisome condition.

## 4. Materials and Methods

### 4.1. Study Design and Cohort

As part of the SepsisDataNet.NRW study (German Clinical Trial Register No. DRKS00018871; http://www.sepsisdatanet.nrw, accessed on 10 January 2024), patients who fulfilled the SEPSIS-3 criteria were prospectively enrolled in a multicenter approach. This study was approved by the Ethics Committee of the Medical Faculty of the Ruhr University Bochum (register number 5047-14). The ethics committees responsible for the respective study centers followed the vote of the Bochum Ethics Committee. This study complied with the revised Declaration of Helsinki, the guidelines for good clinical practice, and local legal requirements. Patients were recruited after providing written informed consent over a period from 1 March 2018 to 31 December 2022 in seven different intensive care units at university hospitals and tertiary hospitals in North Rhine-Westphalia. Adult patients with a sepsis diagnosis within the previous 36 h, according to the current SEPSIS-3 definition (suspected/proven infection and increase in Sequential Organ Failure Assessment (SOFA) score by two points or more), were eligible to participate.

Exclusion criteria were as follows:-Age below 18 years at the time of ICU admission;-Withdrawal or withholding of consent;-Withdrawal of treatment;-COVID-19 positive.

### 4.2. Collection of Blood Samples

Whole blood was drawn at enrolment (day 1) and at day 8 and directly transferred to the laboratory for further processing. Serum was collected after centrifugation of whole blood samples collected in S-Monovetten (Sarstedt, Nümbrecht, Germany). Upon centrifugation (4000× *g*) for 2 min, the serum fraction was removed and stored at −80 °C for cytokine quantification. RNA was collected in Tempus Blood RNA tubes (Applied Biosystems, Waltham, MA, USA) and isolated using a Tempus™ Spin RNA Isolation Kit (Applied Biosystems).

### 4.3. Clinical Data

Medical data, including laboratory values, vital signs, demographic data, point-of-care diagnostics, and length of stay in the intensive care unit, were stored in a comprehensive database (CentraXX 4.0 software, Kairos GmbH, Bochum, Germany) and pseudonymized in accordance with the ethics committee’s requirements.

### 4.4. RNA Quantification, cDNA Synthesis and qPCR

In this study, 1 µg total RNA, which was quantified by spectrometry (260 nm/280 nm), was utilized for cDNA synthesis using a High-Capacity cDNA Reverse Transcription Kit (Thermo Fisher Scientific, Wilmington, NC, USA). qPCR reaction was performed utilizing AQP3 and AQP9 primer (Table 3) and ACTB primer as reference genes, as described previously [45,47].

### 4.5. Measurement of Cytokines in Serum

Serum samples were used to quantify the concentration of thirteen cytokines at the time of recruitment. The LegendPlex Human Inflammation Panel 1 (BioLegend, San Diego, CA, USA) was used according to the manufacturer’s instructions. For this purpose, 25 µL of serum sample was incubated with LegendPlex beads for antigen capture, washed, and incubated with detection antibodies. Fluorescence was then quantified in a flow cytometer (Canto II, BD Biosciences, San Jose, CA, USA). If the measured concentration of a cytokine was outside the standard curve, the following procedure was followed: if it was below the lower limit of detection (LOD), the value was set to 0 pg/mL; if it was above the upper LOD, it was set to the upper LOD.

### 4.6. Statistical Analysis

Patient characteristics are expressed as percentages for categorical variables and as means with SD or medians with interquartile ranges (25th and 75th percentiles). Categorical variables were compared using McNemar’s or Fisher’s exact tests. Continuous independent variables were compared using Student’s t-test or Mann–Whitney U-test. The 30-day survival was compared between groups using a Kaplan–Meier analysis and the log-rank test. The independence of the risk factors was assessed using a Cox regression analysis. The AQP mRNA expression thresholds for the Kaplan–Meier survival analysis were determined via receiver operator characteristic (ROC) curve analysis. The Youden index as point of best discrimination was assessed and then applied in the Kaplan–Meier analysis. The log-rank test was used to determine the statistical significance of the observed survival effect. Hazard ratio was determined using univariate Cox regression analysis. We evaluated the independence of the AQP effect on survival from age and gender using a multivariate Cox regression analysis. A *p*-value of lower than 5% was considered significant. If not stated otherwise, data are always depicted as mean ± standard deviation (SD). All analyses were performed using SPSS (version 28, IBM, Chicago, IL, USA). GraphPad Prism 9 (Graph-Pad, San Diego, CA, USA) was used for graphical presentations.

## Figures and Tables

**Figure 1 ijms-25-01209-f001:**
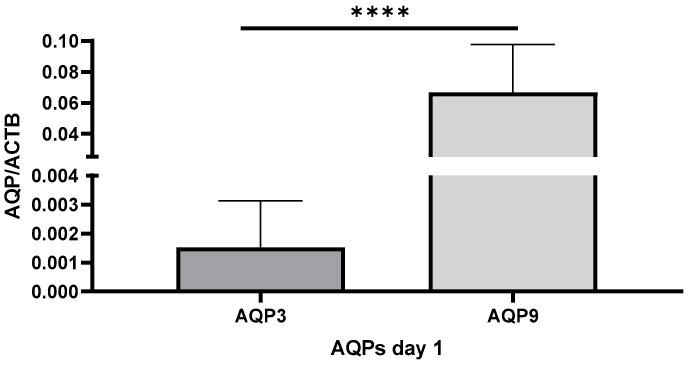
mRNA Expression of AQP3 and AQP9 in the whole blood samples of septic patients, measured by qRT-PCR. AQP3 and AQP9 mRNA was measured in cDNA samples relative to the housekeeping gene β-actin (ACTB). *n* = 87, unpaired *t*-Test, **** *p* < 0.0001.

**Figure 2 ijms-25-01209-f002:**
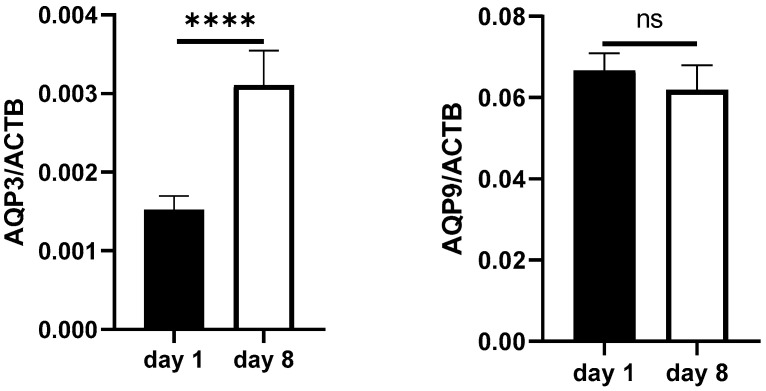
mRNA AQP expression in the whole blood samples of septic patients on day 1 and day 8 after sepsis diagnosis measured by qRT-PCR. AQP3 and AQP9 mRNA were measured in cDNA samples relative to the housekeeping gene β-actin (ACTB). *n* = 87 (day 1); *n* = 36 (day 8), Mann–Whitney test, **** *p* < 0.0001, ns = not significant.

**Figure 3 ijms-25-01209-f003:**
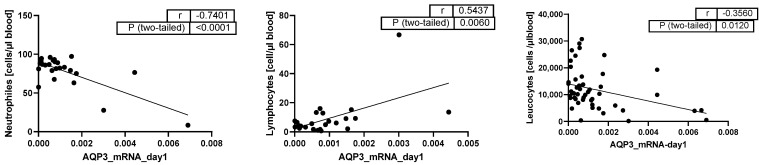
Pearson Correlation analysis of AQP3 mRNA expression with cell counts (cells per µL blood) (*n* = 24 neutrophils; *n* = 25 lymphocytes; and *n* = 49 leucocytes); dots: the two variables of the same patient, line: correlation line of all patients.

**Figure 4 ijms-25-01209-f004:**
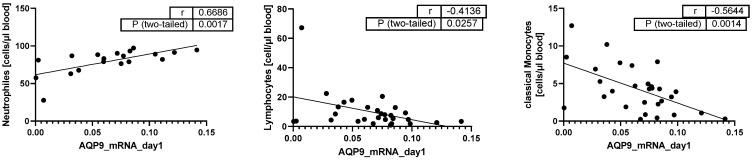
Pearson Correlation analysis of AQP9 mRNA expression with cell counts (cells per µL blood) (*n* = 19 neutrophils; *n* = 29 lymphocytes; and *n* = 29 classical monocytes); dots: the two variables of the same patient, line: correlation line of all patients.

**Figure 5 ijms-25-01209-f005:**
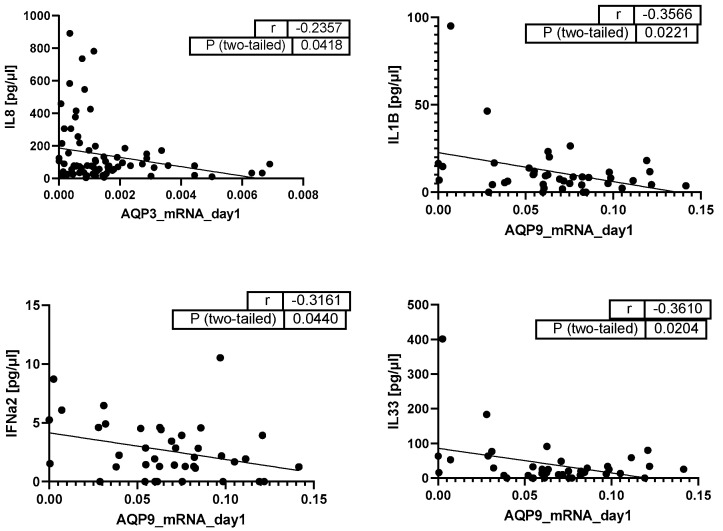
Pearson Correlation analysis of AQP3 and AQP9 mRNA with cytokines on day 1 (pg/µL) (*n* = 75 for correlations with AQP3 and *n* = 41 for correlations with AQP9); dots: the two variables of the same patient, line: correlation line of all patients.

**Figure 6 ijms-25-01209-f006:**
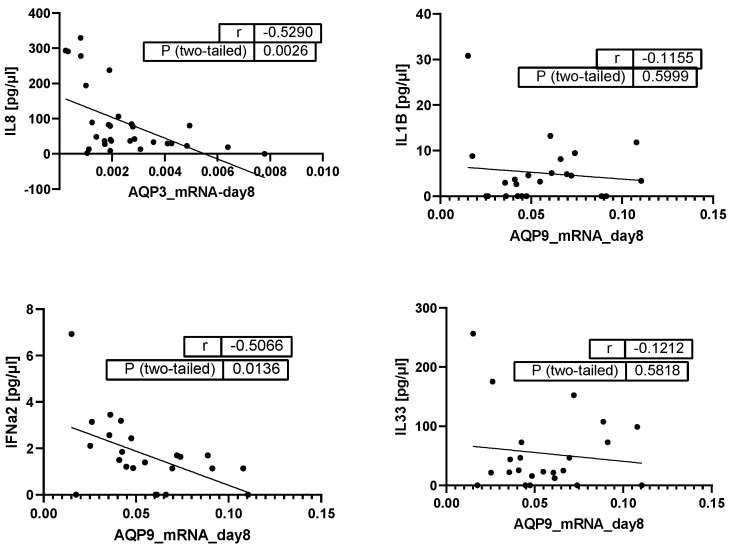
Pearson Correlation analysis of AQP3 and AQP9 mRNA with cytokines on day 8 (pg/µL) (*n* = 30 for correlations with AQP3 and *n* = 23 for correlations with AQP9); dots: the two variables of the same patient, line: correlation line of all patients.

**Figure 7 ijms-25-01209-f007:**
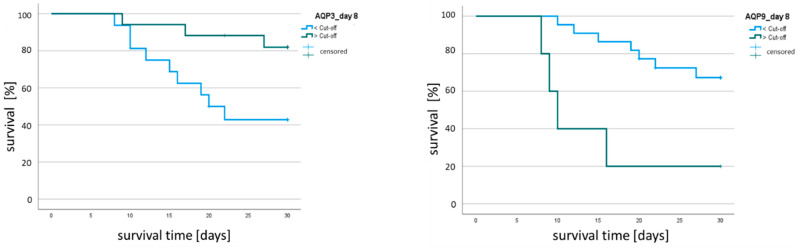
Kaplan–Meier analysis of 30-day sepsis survival, with septic patients stratified using cut-off values of AQP3 and AQP9 mRNA expression relative to ACTB. Cut-off values were determined using ROC analysis and the Youden index. The patient cohort was divided by cut-off value: the cut-off for AQP3 was 0.00223 (*n* = 33), and that for AQP9 was 0.088078 (*n* = 27). Green line: patients exceeding the cut-off; blue line: patients falling below the cut-off.

**Table 1 ijms-25-01209-t001:** Baseline characteristics of the sepsis patient cohort.

	Entire Cohort	*n*
*n*	87	87
Male gender, *n* (%)	46 (52.9%)	87
Age years, median (IQR)	68 (61–79)	81
SOFA score, median (IQR)	8 (5–11)	82
SAPS-2 DRG, median (IQR)	36 (27–44)	65
PCT ng/mL, median (IQR)	3.9 (0.4–12.2)	57
CRP mg/dl, median (IQR)	16 (9.7–24.6)	57
Lactate mM, median (IQR)	1.4 (0.9–2.4)	66
Comorbidities, *n* (%)	87	87
- Alcohol	6 (6.9%)	
- Chronic kidney disease	19 (21.8%)	
- Hypertension	47 (54.0%)	
- Diabetes	19 (21.8%)	
- Obesity	44 (16.1%)	
- Cardiovascular	27 (31.0%)	
- Malignancies	11 (12.6%)	
- Nicotine	11 (12.6%)	
- Dialysis	5 (5.7%)	
- Transplantation	14 (16.1%)	
- COPD	12 (13.8%)	
- Other (lungs)	8 (9.2%)	
Focus of infection, *n* (%)	79	79
- Central nervous system	2 (2.5%)	
- Lower respiratory tract	34 (43.0%)	
- Skin and soft tissue	5 (6.3%)	
- Genitourinary	9 (11.4%)	
- Cardiovascular	5 (6.3%)	
- Intra-abdominal	21 (26.6%)	
- Musculoskeletal	1 (1.3%)	
ICU length of stay, median days (IQR)	7 (3–13)	76
Hospital length of stay, median days (IQR)	15 (7–28)	52
30-day mortality, *n* (%)	30 (34.5%)	87

**Table 2 ijms-25-01209-t002:** Multivariate Cox regression of the AQP9 cut-off.

	Hazard Ratio	*p*-Value	95% Confidence Interval
AQP9_mRNA_day8_cut-off	10.285	0.003	2.268–46.640
Age	0.736	0.646	0.199–2.719
gender	1.051	0.023	1.007–1.096

**Table 3 ijms-25-01209-t003:** Primer sequences.

Primer Name	Sequence
AQP1_Rt_Se	5′-GTAGCCAGCACGCATAGCAC-3′
AQP1_Rt_As	5′-GCCATCCTCTCAGGCATCAC-3′
AQP3_Se	5′-GGAATAGTTTTTGGGCTGTA-3′
AQP3_As	5′-GGCTGTGCCTATGAACTGGT-3′
AQP5_Rt_SE	5′-TCCATTGGCCTGTCTGTCAC-3′
AQP5_Rt_AS	5′-ACCCAGAAAACCCAGTGAGC-3′
AQP7_Rt2_Se	5′-GGACTGGGGACACAGGGATA-3′
AQP7_Rt2_As	5′-GCTGAAAGTGCAATCCACGG-3′
AQP8_RT-SE	5′-GAGATCATCCTGACGACGCT-3′
AQP8_RT-AS	5′-TTCATGCAGCCTCCAGACAC-3′
AQP9_RT_Se	5′-ATGTGGGAGCCCAGTTCTTG-3′
AQP9_Rt-As:	5′-TACGGAGCTGGGTATGTTGC-3′
AQP10_Rt_Se	5′-CTACGTGGGTGGTAACGTCTC-3′
AQP10_Rt_As	5′-TAGGTGGCAAAAATGGAGGCT-3′

## Data Availability

The data presented in this study are available upon request from the corresponding author.

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
