# Peer review of "AQP3 and AQP9—Contrary Players in Sepsis?"

_ijms, 2024, doi:10.3390/ijms25021209_

Round 1
Reviewer 1 Report
Comments and Suggestions for Authors
The current manuscript entitled, “AQP3 and AQP9-contrary players in sepsis?” deals with the important regulatory roles of aquaporins as a key player in sepsis. There are certain areas which could have been focused well. Few recommendations are given below:
Comments:
1. The introduction is short and doesn’t cover the topic at large. It must provide relevant information on global burden of sepsis related mortality and morbidity. Biomarkers, diagnosis of sepsis, and sepsis therapeutics could be highlighted.
2. Studies suggests that the COVID-19 virus was a more common and deadly cause of sepsis early. Since, the sample collection was from March 2018- Dec 2022. Did this criterion was considered as the post-covid, patients develop more severe sepsis which is also known to alter the AQP expressions as well.
3. Why were the healthy controls not taken up for the study? Also, there is no mention about the severity of sepsis in patient samples.
4. Figure legends needs improvements as necessary information is not provided.
5. Strong evidence showed that AQP1, AQP3, AQP5, AQP7 and AQP9 have been involved in sepsis. Why the authors specifically targeted AQP1 and AQP9? Why others aquaporins were not considered for the study? It would be better to specifically understand the regulatory role of others too.
6. It would be better if the authors would have done the whole blood transcriptome sequencing data of the patients to examine the alteration of immune cell types, expression pattern of cell death signaling genes, as well as differentially expressed lncRNAs.
7. Why were the cytokines levels not shown for Day 8? Did the authors find any changes? It would be important to highlight both time points.
8. IL-33 is more of the events of ILC2 population related inflammatory responses. Did the authors check these populations?
Author Response
Thank you very much for reading our manuscript. They have greatly improved it through your work

Reviewer 2 Report
Comments and Suggestions for Authors
This article studied the relationship of AQP3 and AQP9 on sepsis using patients’ samples, which has high clinical significance. However, there are some minor shortcomings.
1. Authors should describe the function of cells (neutrophiles, lymphocytes, leucocytes, classic monocytes, etc.) and cytokines (IL-8, IL-1B, IFN-α2, IL-33 ) on sepsis, and discuss the relationship with AQP3 and AQP9.
2. In line 90-92 “AQP3 expression correlated negative with neutrophil (p < 0.0001, figure 4) and leucocyte cell count (p=0.012, figure 4) and positive with lymphocytes cell count (p = 0.006, figure 3)”, line 95-97 “In contrast to AQP3, AQP9 showed a strong positive correlation with neutrophils cell 95 count (p = 0.0017, figure 5) and a negative correlation with lymphocytes and classical 96 monocytes (p < 0.025; figure 4)”, the description isn’t consistent with the figure.
Author Response
Thank you very much for reviewing our manuscript.

Round 2
Reviewer 1 Report
Comments and Suggestions for Authors
The authors have done a fantastic job of improving the quality and scientific rigor of the study. All of the comments are taken care of and necessary information, including data, is added to the modified version of the manuscript. There is overall improvement that can be suitable for publication.